


**Modelling the compound flood hydrodynamics under mesh convergence and**
**future storm surge events in Brisbane River Estuary, Australia**
Usman Khalil [1,*], Shuqing Yang [1], Muttucumaru Sivakumar [1], Keith Enever [1], Muhammad
Zain Bin Riaz [1], Mariam Sajid [2]
[1] School of Civil, Mining and Environmental Engineering, University of Wollongong, NSW
2522, Australia
[2] School of Architecture and Design, University of Engineering and Technology Lahore, 54890,
Lahore Pakistan
* Corresponding Author: Usman Khalil (uk998@uowmail.edu.au)



**Abstract:** Floods are the most common and destructive disasters around the globe, which becomes
more challenging in coastal areas due to higher population density and catchment area relative to
floods in an inland area. For effective coastal flood management to reduce flood adverse impacts it is
necessary to investigate the flooding processes and their joint interaction in a coastal area. This paper
selected the Brisbane River Estuary, Australia as an example and the MIKE 21 model is applied to
investigate the effects of mesh resolution on the flood discharge and to explore compound flooding
by computing variances in coastal flood assessments resulting from a separation of tidal and riverine
processes. The statistical results showed that the Nash-Sutcliffe coefficient, E of water level are varied
from 0.84 to 0.95 and the model simulated the 2011 flood extent results agreed with 90% accuracy
with the observed flood extent. Five mesh resolutions cases were analyzed and the result found that
the finer mesh resolution Case 5 was more appropriate for calculating the peak discharge with 2.7%
with estimated discharge. Compound flood event simulation results emphasized that not considering
the interaction of various flooding drivers caused 0.62 m and 0.12 m reduction in the flood levels at
Jindalee and Brisbane city gauges, and uncertainties in flood extent. Simulated results of flood at
Brisbane city gauge, showed that 2011 and 2013 floods with storm surge scenario 4 demonstrate, the
increase in flood level to be 12% and 34% respectively. The results recommend flooding assessment
by using mesh convergence with joint probability of compound flood under future storm surge for
planning and management of coastal projects.
**Keywords:** Compound flood, hydrodynamic model, MIKE 21, Mesh size, storm surge
**1   Introduction**
Flooding is a prevalent and most destructive catastrophe worldwide, which poses a severe threat to
lives and properties (Geravand, Hosseini, & Ataie-Ashtiani, 2020; Khalil & Khan, 2017; X. Liu &
Lim, 2017). Coastal flooding is likely to increase in the future (Sadler, Goodall, Behl, Bowes, &

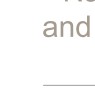
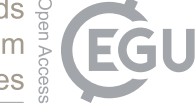

Morsy, 2020) due to sea-level rise, increased storm surge, land subsidence and urbanization
(Pachauri et al., 2014; Van Coppenolle & Temmerman, 2019; Vitousek et al., 2017). Flooding has
caused a global economic loss of $US 70.1 billion between 2000-2015 (Geravand et al., 2020),
affecting 2.3 billion people (Hallegatte, Green, Nicholls, & Corfee-Morlot, 2013). It is anticipated
that global flood losses will hit USD one trillion per year in 2050 (Lee & Kim, 2018; McCallum et
al., 2016; Sulis, Frongia, Liberatore, Zucca, & Sechi, 2018; Tsoukala et al., 2016; UNISDR, 2015).
For instance, in South East Queensland (SEQ), Australia the 2011 flood affected more than 2.5
million people and around 29,000 homes in the Brisbane River Valley (Barton, Wallace, Syme,
Wong, & Onta, 2015; Syme, Wallace, Rodgers, Jensen, & Barton, 2019), the flooding led to 35
deaths and AUD2.55 billion economic loss (van den Honert & McAneney, 2011). In coastal
catchments, floods can be produced by runoff generated by a significant rainfall event (Neumann et
al., 2014) and a raised ocean level produced by a storm surge, or a combination of both. A storm
surge is the rise of water level above the normal sea level along a coast due to reduced atmospheric
pressure and/or strong coastal winds (Karim & Mimura, 2008). The storm surge influences may
further increase when they coincide with riverine flooding (Zheng, Westra, Leonard, & Sisson,
2014) and the resulting combination is known as compound flood events (Leonard et al., 2014; Wu,
Westra, & Leonard, 2020). Initially, the two involved flooding drivers involved were managed
individually in coastal flood management (Torres et al., 2015). However, studies show that storm
surge and extreme rainfall processes are statistically dependent, and thus their joint interface needs
to be considered (Hawkes & Svensson, 2006; Svensson & Jones, 2004; Zheng, Westra, & Sisson,
2013). For an effective coastal flood inundation assessment, multiple measures, such as flood
assessment models using good resolution digital elevation data (DEM) and considering a compound
flood event with future storm surges are required to be implemented to reduce flood adverse impacts.
In the past various researchers have given substantial efforts to simulate flood inundation in the



coastal floodplain with different numerical modelling approaches (Chen, Evans, Djordjević, &
Savić, 2012; Son, Kim, & Han, 2016). Modelling approaches considered were (1) empirical
approaches such as measurements (O'Connor & Costa, 2004) and remote sensing (Smith, 1997), (2)
hydrodynamic models; (3) conceptual models for big floodplain areas (Teng et al., 2017) and
probabilistic flood risk assessment (Apel, Thieken, Merz, Blöschl, & Sciences, 2004). While, the
hydrodynamic models and in particular 2D models were the most used tools to simulate flood
hydrodynamics, flood extent, flood forecasting and scenario analysis (Teng et al., 2017) because
they simulate features in x and y directions at every mesh point interval, and take fewer computation
times as compared to 3D model (X. Liu & Lim, 2017; Shrestha et al., 2020; Yu, 2017). Further,
flood hydrodynamic modelling has significantly improved with the advanced approaches including
digital elevation models (DEMs). Flood modelling simulation accuracy mainly depends on the
suitable discretization of the geometric domain and grid resolution of the DEM (Geravand et al.,
2020). Literature review revealed that existing studies have used various resolution DEM data with
and without considering the compound flood and future storm surge events effects (Kumbier, Cabral
Carvalho, Vafeidis, & Woodroffe, 2018). For instance, Karim and Mimura (2008) studied the
influences of sea-level rise (SLR) on storm surge flooding in Western Bangladesh and
hydrodynamic model simulation showed that for a storm surge under 0.3m SLR, the flooded area
would enlarge by 15.3% of the current flooded area. Similarly, studies showed that higher resolution
DEMs represents more exact terrain features which leads to more precise floodplain assessment and
coarser-resolution DEMs over-predicted the extents of flood and caused substantial precision loss
(Shrestha et al., 2020). However, there is no systematic framework for considering all the relevant
flooding modelling parameters for effective flood assessment.
The recent flooding in summer 2021 has caused widespread damages across 20 countries (i.e. the
USA to Italy, from China to India etc.) with 920 deaths (Copernicus, 2021). Further, climate change





impacts assessments in IPCC recent report (IPCC, 2021) have urged us to consider and model future
impacts and take remedial measures to reduce their impacts. Flooding events specify the need for a
sustainable modelling approach to simulate the flood extent and propose measures to alleviate future
flooding. To access the flood hydrodynamics, the Brisbane River Estuary (BRE) Australia were
studied, which has been exposed to several flooding events over the past century, while, it has
experienced two destructive flooding events in the year 2011 and 2013 when a storm surge
coincided with extreme riverine discharge in the BRE (Queensland Floods Commission Inquiry
Report, 2011). Earlier studies have utilized the 2D hydrodynamic model for Brisbane river flood
modelling (X. Liu & Lim, 2017; Yu, 2017). However, the combined effect of riverine and tidal
flooding was not considered, with the future effect of storm surge (Wu et al., 2020). For flood
assessment and inundation mapping both temporal and spatial (flood depth and inundation extent)
knowledge is required and can be applied in the flood risk analysis (X. Liu & Lim, 2017; Mani,
Chatterjee, & Kumar, 2014). However, existing studies on the Brisbane River have used coarser-
resolution geometric data, with 66.9% accuracy of flood modelling results, which leads to
uncertainty and is less precise for coastal management. Other studies used hydrological and
hydraulic models combination for BRE flood extent assessment, however, these models require
extensive input data and further, it is restricted to government use and have limited access for
research studies (Barton, Syme, Ryan, Rodgers, & Jensen, 2017). The complexity of the Brisbane
river area for flood forecast by analyzing the compound flooding risks and considering future storm
surge scenarios is still a significant and challenging study (Pellikka, Leijala, Johansson, Leinonen,
& Kahma, 2018), which lead to the development of the hydrodynamic model for BRE for flood
inundation.
This study examines and analyses BRE hydrodynamics by using MIKE 21 hydrodynamic model to
investigate the flood extent with various mesh resolutions. Further, to understand the interaction of

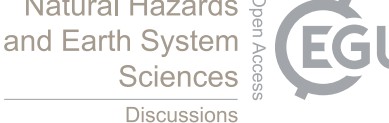

storm-tide and fluvial flooding mechanisms, we investigate a compound flood event in a BRE to
quantify how changing boundary set-ups at the entrance in Moreton Bay affect modelled water
levels and flood extent of the study area. In this paper, we address several of the issues outlined
above; the objectives of this study are threefold: (1) to develop a Brisbane River estuary and
Moreton Bay flood model for various flows events under converging mesh size; (2) to conduct a
simulation to assess the impact of compound riverine flooding and tides on water levels; and (3) to
analyse the future storm surge effect on flood extent. The outcome of this study will be significant
to comprehend the suitability of the hydrodynamic model to carry out flood modelling. Further, it
will help to identify the flood-exposed areas and to apply possible remedial strategies to overcome
the damage. The study results will help decision-makers to make a flood mitigation and management
plan of action.
The paper is structured as follows: the case study area, Brisbane River estuary (BRE) is described
in Section 2. Section 3 explains the Brisbane flooding, hydrodynamic model, the data requirements,
and methods, including mesh resolution effect and compound flooding in BRE. The results of the
hydrodynamic model calibration and validation along with mesh effects and compound flooding
influence are described and discussed in Section 4. Finally, conclusions are specified in Section 5.

## 124 2   Study area

The Brisbane River and Moreton Bay are located on the southeast coast of Queensland, Australia
(Fig. 1). The lower part of Brisbane River is termed the Brisbane River estuary (BRE). Moreton
Bay is semi-closed coastal water situated at the mouth of Brisbane River. BRE and Moreton bay
experience semidiurnal tides with a tidal range of 2.5 m. The Brisbane River has the longest course
in sub-tropical SEQ, having a length of 344 km and has a catchment area of 13,600 km$^2$ (Eyre,
Hossain, & McKee, 1998) to the Port Office Gauge which is located in the heart of Brisbane City.



The BRE is a micro-tidal estuary, with a mean spring and neap tidal range of 1.8 m and 1.0 m
respectively (Wolanski, 2014). It has a tidal influence up to 80 km from the river mouth. The Oxley
Creek and Bremer Rivers are major tributaries, which contribute to lower half catchment flows in
BRE and join the estuary at 34 and 73 km respectively, from the river mouth. Estuary depth varies,
from 15 m at the river mouth to about 4 m above Australian Height Datum (AHD) at the Bremer
River junction at Moggill Point.
The catchment is manifold, joining rural and urban land, dams for flood mitigations, tidal impacts,
and various tributaries with the prospect of flooding. The river system itself contains the Brisbane
River and numerous main tributaries. The Brisbane River has two dams situated in its upper
catchment, both of which were constructed with the twin objective of flood alleviation and water
supply to Brisbane City. Wivenhoe dam regulates the flow of water in the upper part of Brisbane
River, which is approximately 150 km upstream of the coast. The annual mean rainfall of the
Brisbane River catchment is around 990 mm per year. In January 2011, a storm event caused
widespread inundation on BRE floodplains (van den Honert & McAneney, 2011). Further, the 2013
storm and tidal influence caused mild inundation in BRE.

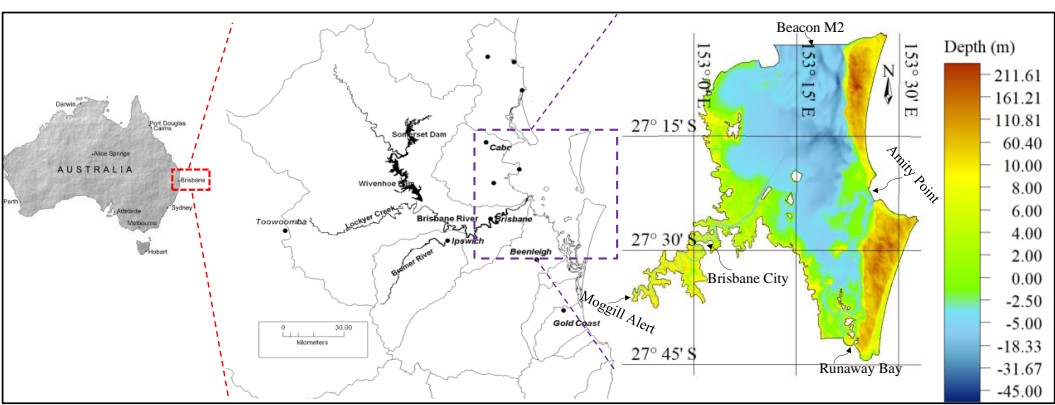


Fig 1: Brisbane River catchment and Moreton Bay


## 3    Materials and methods


This study utilizes the MIKE 21 FM to explore the hydrodynamics and flood inundation in BRE.
The MIKE-21 hydrodynamic model is built with various fine resolution mesh data and a time series
of observed water levels and discharges were used to force model boundaries. Observed flood extent
imagery, tidal gauges and water level were used for modelling results validation. By modelling the
involved flooding drivers individually and jointly, we compute differences in flood risk estimation
resulting from a separation of riverine and tide processes in coastal flood modelling. The modelling
process, calibration and validation of the model followed in this study is presented in a flow chart
as shown in Fig. 2.

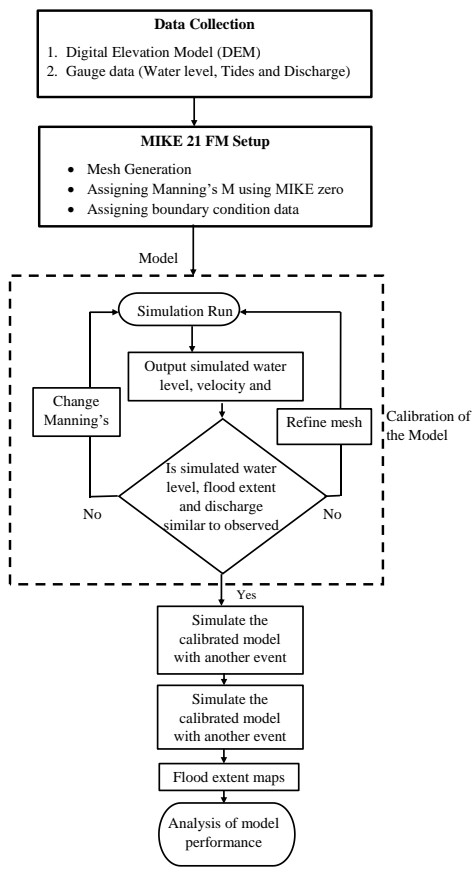


Fig. 2: Flowchart of processes involved in the MIKE 21 hydrodynamic simulation


*3.1 Data collection*
To carry out this study we have collected, DEM bathymetry data, water level, flow data, tidal data,
flood measurement data, and flood extent satellite data. Data collected for the study with its
resolution and sources are shown in Table 1.
Table 1. Data collected and sources for the study

| Data | Description | Station No | Sources |
|---|---|---|---|
| DEM data | Brisbane River and floodplains; 5m x 5m DEM | N/A | Queensland Government under the Brisbane River Catchment Flood Study (BRCFS) |
| | Moreton Bay; 30m x 30m DEM | N/A | James Cook University (JCU) 3DGBR: GBR100 High-resolution |
| River gauge data (instantaneous values) | Moggill Alert | 540200 | Bureau of Meteorology, Queensland |
| | Jindalee | 540192 | |
| | Brisbane City Alert | 540198 | |
| Flow data (daily) | Savage crossing | 143001 | Water Monitoring Information Portal, Queensland Government |
| Tidal data (10 minutes interval) | Beacon M2 Moreton Bay | 046206A | Maritime Safety Queensland |
| | Runaway Bay | 045100B | |
| | Amity Point | 046211E | |
| | Brisbane Bar | 046046A | |
| | Whyte Island Alert | 540495 | |
| Flood extent (2011 and 2013) | Brisbane River flood extent | N/A | Queensland Government, Flood imagery and data |


*3.2 Hydrodynamic model*
The model used in this study is DHI MIKE-21 FM, which is a two-dimensional (2D), depth-
averaged hydrodynamic model, with a numerical solution based on the incompressible Reynolds
averaged Navier-Stokes equations while using the finite volume method to solve the shallow water
equations (DHI, 2017a). In the shallow water hydrodynamic equations, due to the stability constraint
of the explicit scheme, the Courant-Friedrich-Levy (CFL) condition needs to be fulfilled, which can
be calculated in Eq. 1. Critical CFL values are recommended to be set at 0.8, to fully secure the
stability of the numerical scheme (DHI, 2014)




$$CFL_{HD} = \left(\sqrt{gh} + |u|\right)\frac{\Delta t}{\Delta x} + \left(\sqrt{gh} + |v|\right)\frac{\Delta t}{\Delta y} \tag{1}$$

Where h, is the local water depth; $\Delta t$ is the interval of time step; $\Delta x$ and $\Delta y$ are typical length scales
of meshes in the x and y direction respectively, u and v are the velocity components. The governing
equation may not account for the wrong topography representation and its subsequent errors in the
results. The correct discretization of mesh elements and limiting CFL can lead to correct results.
MIKE 21 governing equations are attained by the integration of the horizontal momentum and
continuity equation over depth $h = \eta + d$; the following shallow water 2-D equations are defined in
Eqs (1-3):

$$\frac{\partial h}{\partial t} + \frac{\partial h\bar{u}}{\partial x} + \frac{\partial h\bar{v}}{\partial y} = hS \tag{2}$$

$$\frac{\partial h\bar{u}}{\partial t} + \frac{\partial h\bar{u}^2}{\partial x} + \frac{\partial h\bar{v}\bar{u}}{\partial y}$$

$$= f\bar{v}h - gh\frac{\partial \eta}{\partial x} - \frac{h}{\rho_o}\frac{\partial P_a}{\partial x} - \frac{gh^2}{2\rho_o}\frac{\partial \rho}{\partial x} + \frac{\tau_{sx}}{\rho_o} + \frac{\tau_{bx}}{\rho_o} - \frac{1}{\rho_o}\left(\frac{\partial s_{xx}}{\partial x} + \frac{\partial s_{xy}}{\partial y}\right) \tag{3}$$

$$+ \frac{\partial (hT_{xx})}{\partial x} + \frac{\partial (hT_{xy})}{\partial y} + hu_s$$

$$\frac{\partial h\bar{v}}{\partial t} + \frac{\partial h\bar{v}\bar{u}}{\partial x} + \frac{\partial hv^2}{\partial y}$$

$$= f\bar{u}h - gh\frac{\partial \eta}{\partial y} - \frac{h}{\rho_o}\frac{\partial P_a}{\partial y} - \frac{gh^2}{2\rho_o}\frac{\partial \rho}{\partial y} + \frac{\tau_{sy}}{\rho_o} + \frac{\tau_{by}}{\rho_o} - \frac{1}{\rho_o}\left(\frac{\partial s_{yx}}{\partial x} + \frac{\partial s_{yy}}{\partial y}\right) \tag{4}$$

$$+ \frac{\partial (hT_{xy})}{\partial x} + \frac{\partial (hT_{yy})}{\partial y} + hv_s S$$

Where t is the time, $x$ and y are Cartesian coordinates,  h is the water depth, s is the discharge, $\rho_o$ is
reference density of water, $\rho$ is water density, $P_a$ is atmospheric pressure (P$_a$), $s_{xx}$ , $s_{yx}$ , $s_{yy}$ , $s_{xy}$
are radiation stress components, $\bar{v}$ , $\bar{u}$ are y and x directions depth-averaged velocity, $\tau_{xx}$ , $\tau_{xy}$ , $\tau_{yx}$,



$\tau_{yy}$ are lateral stress components, $\tau_{sx}$, $\tau_{sy}$ are surface wind stress components, g is the acceleration
due to gravity, f is Coriolis parameter, $v_s$, $u_s$ are velocity by which water is discharged into the
ambient water, $\tau_{bx}$, $\tau_{by}$ are bottom stress components and $\eta$ is water surface elevation.
The Flow Model of MIKE 21 is the basic hydrodynamic module. It provides the hydrodynamic
basis for the computations of coastal hydraulics. It models the flows and variations of water level in
response to a range of forcing functions on floodplains, lakes, estuaries, and coastal areas. Many
researchers have successfully used the MIKE 21 FM model to investigate the hydrodynamic process
of a large river and coastal bay, like Dongting Lake, China (Y. Liu et al., 2019) the Poyang Lake,
China (Li & Yao, 2015; Li, Zhang, Tan, & Yao, 2020), Brisbane River estuary (Khalil et al., 2020),
Vembanad Lake, India (Haldar, Khosa, & Gosain, 2019), Lake Alexandrina (J. Liu, Sivakumar,
Yang, & Jones, 2018) and Deer Creek in the City of Brentwood (Shrestha et al., 2020). The MIKE
21 FM hydrodynamic module was used to simulate the depth-averaged flow features for the years
2006, 2011 and 2013 of the Brisbane River and Moreton Bay.
*3.3    Mesh generation and cell size convergence*
The DHI MIKE mesh generation tool allows the user to design the element resolution by describing
the maximum element area, $A_{max}$. For the majority of mesh structure elements, approximate
equilateral triangles can be attained in DHI MIKE (DHI, 2012) and the length of $\Delta x$ is
approximately calculated using Eq. 5, where $\theta \approx 60$;
$$\Delta x_{max} = 2\sqrt{A_{max} \cdot \tan(\theta/2)} \tag{5}$$
In $CFL_{HD}$ Eq. 1, as the local flow velocity is very less as compared to the local water depth so, it is
reasonable to disregard the velocity terms, and the $CFL_{HD}$ can be rewritten as Eq. 6 and further, the
$CFL_{HD}$ is rearranged to Eq. 7 with the reasonable assumption of $\Delta x \approx \Delta y$.


$$CFL_{HD} = \sqrt{gh}\frac{\Delta t}{\Delta x} + \sqrt{gh}\frac{\Delta t}{\Delta y} \qquad\qquad (6)$$
$$CFL_{HD} = \sqrt{gh}\frac{\Delta t}{\sqrt{A_{max}\,.\tan(\theta/2)}} \leq 0.8 \qquad\qquad (7)$$
Five meshes were generated and gradually attuned until all elements fulfilled the constraint in Eq.
7. Mesh quality was further enhanced by the smoothing tool to increase spatial regularity. The
details of the elements, nodes, element areas $CFL_{max}$ and simulation running time in each case are
given in Table 2. For the case 1 mesh file, the $A_{max}$ is 5 km$^2$ and $A_{min}$ is 70 m$^2$, generating 95,497
elements covering the Brisbane River estuary with floodplain and the entire Moreton Bay. The mesh
sizes were reduced in each subsequent case and in case 5 the mesh elements were 288,415. The five
mesh cases with different element sizes for a small region near Brisbane city are shown in Fig. 3. In
these five cases, element sizes were distributed with finer elements inside the Brisbane River and
coarser elements inside Moreton Bay. In all cases, the time step, $\Delta t$, was set as 30 seconds to fulfil
the critical $CFL_{HD}$ of 0.8. A larger number of elements were estimated to involve a much lengthier
time to complete the simulations. For the one month simulation of the flood event in BRE and
Moreton Bay, the running time was approximately 11, 13, 18, 28 and 36 hours for each case,
respectively.
Table. 2. Statistics of mesh convergence cases

| | Case 1 | Case 2 | Case 3 | Case 4 | Case 5 |
|---|---|---|---|---|---|
| Number of elements | 95,497 | 122,220 | 153,383 | 217,066 | 288,415 |
| Number of nodes | 48,215 | 61,605 | 77,352 | 11,3116 | 148,697 |
| Element area max. x10$^4$ (m$^2$) | 500 | 100 | 30 | 15 | 10 |
| Element area min. (m$^2$) | 70 | 59.9 | 46.7 | 27.3 | 21.5 |
| $CFL_{max}$ | 0.53 | 0.57 | 0.65 | 0.74 | 0.79 |
| Running time (Hours) | 11 | 13 | 18 | 28 | 36 |

Cell size convergence was performed until the increase or decree of cell size of the mesh make an
effect on water levels. When the results were consistent, then we used the coarser-resolution with
confidence, otherwise, we reduced the cell size until we got a consistent value.


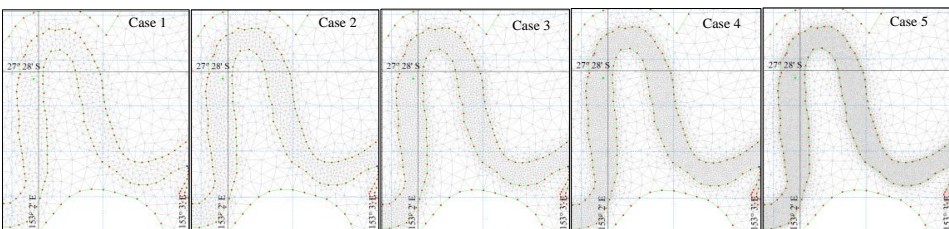


Fig. 3. Five cases of mesh convergence near the Brisbane city used in the modelling

The bathymetric data, longitudinal profile and cross-section comparison are shown in Fig. 4. It can
be seen that by using coarser cell size and with uncontrolled data the cross-section representation
changed as compared to finer cell and controlled data (Fig. 4e)

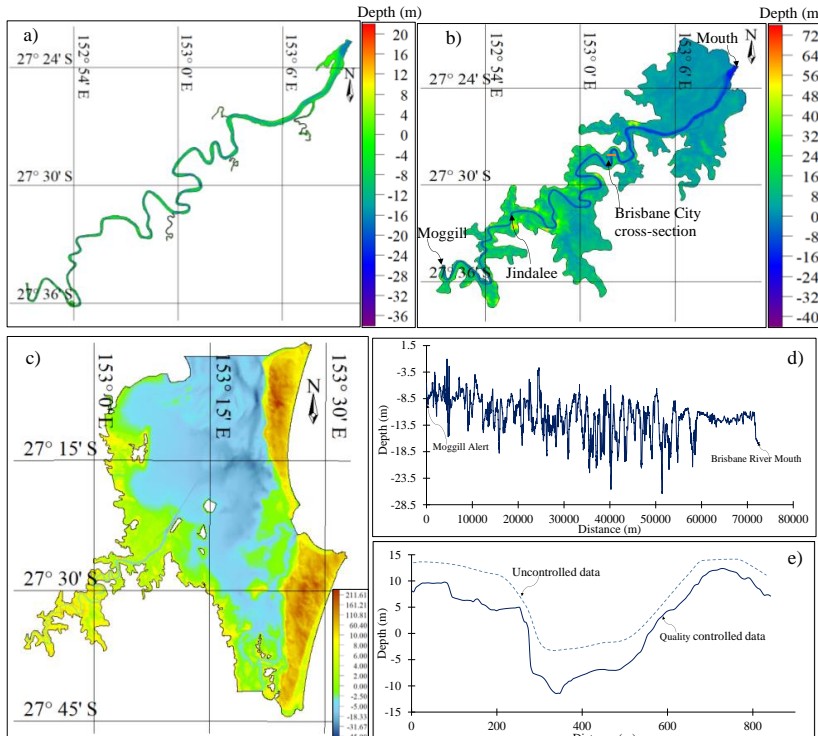


Fig. 4: Bathymetry of a) Brisbane River; b) Brisbane River and floodplain; c) Brisbane river

floodplains and Moreton Bay; d) Longitudinal profile of Brisbane river; e) Brisbane river cross-

section at Brisbane city



*3.4    Future scenarios of storm surge*
Tropical cyclones at some hundred kilometers north of Moreton Bay cause high waves and storm
surge inside Moreton Bay. These events usually do not produce storm winds within Moreton Bay;
however, can produce big ocean swell waves. It has been recognized that the wave circumstances
produced from far away cyclones can cause a deviation of Moreton Bay water levels (Treloar,
Taylor, & Prenzler, 2011). The mixture of storm surge and the normal tide is known as a storm tide
and disastrous impacts occur when the storm surge coincides with an existing high tide. As a result,
the storm tide can have an influence further upstream in the estuary. The ongoing sea-level rise will
cause a rise in river tail-water levels, particularly in the storm surge. As the sea-level rise projections
range is wide enough, while along the Australian coast, sea-level growth maybe 10% higher (IPCC,
2007). Ayre, Diermanse, L, and Hart (2017) provided the cases of flood simulation for four periods
of future climate changes i.e. 2030, 2050, 2070 and 2100, in the BRE which is shown in Table 3.
Based on these four scenarios, storm tide inputs at Brisbane bar (Fig. 5) were used to simulate the
flooding behavior in BRE under low flow events.
Table 3: Proposed climate parameters for inclusion in BRE flood risk study

| Scenario (S) | Years | Storm tide level |
|---|---|---|
| Base Scenario | Present | -1.5 to 1.5m |
| Scenario 1 | 2030 | 25% increase to the base case |
| Scenario 2 | 2050 | 50 % increase to the base case |
| Scenario 3 | 2070 | 75 % increase to the base case |
| Scenario 4 | 2100 | 100 % increase to the base case |

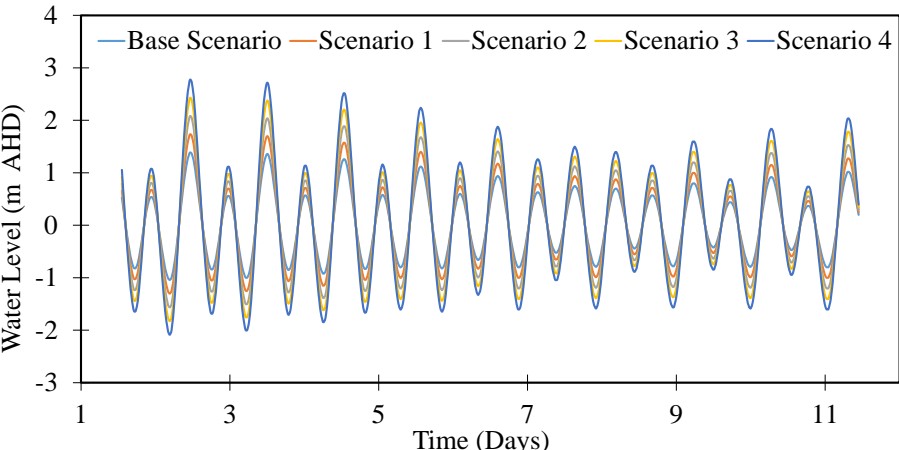


Fig 5: Input data for various scenarios of storm surge tide at Brisbane Bar

*3.5    Model Setup*
In the model, five meshes were generated the element sizes were adjusted by mesh convergence test
and finally a range from 34 m$^2$ to 100,000 m$^2$, with a total of 288,415 unstructured elements and
148,697 nodes were used in the MIKE 21 domain area. The observed time-varying water level was
used as the upstream boundary to the hydrodynamic model, and the tidal level observations at three
stations (i.e., Amity Point, Runaway Bay Point, and at Beacon M2 Point) were adopted to set the
lower boundary condition of the model (Fig ). The model was initially set up by using bathymetric
data of Brisbane River (Fig. 4a) and 2006 low flow data and tidal data as boundary conditions (Fig.
6 a&b). Then the model was extended to floodplain area (Fig. 4b) and the years 2013 and 2011 flood
and tidal data (Fig. 6 c&d) were used for the model performance. Finally, the model included BRE
and Moreton bay (Fig. 4c) with 2011 flood and tidal data (Fig. 6 e-h).
Currently, the MIKE-21 model has not considered evaporation, precipitation, wind direction, and
wind speed. The initial water surface elevation of 0.1 m was used, while the initial water flow
velocities were fixed to zero across the model area. The minimum time step is limited to 0.1 s to




keep the target Courant-Friedrichs-Lewy (CFL) number below 1.0. The thresholds $h_{dry}$ (drying
depth = 0.005 m) < $h_{flood}$ (flooding depth = 0.05 m) < $h_{wet}$ (wetting depth = 0.1 m) were used to
describe the change of wetting and drying in the model (Li et al., 2020). In a hydrodynamic model,
bed resistance is an important factor that controls river flow behaviour. While calibrating the model,
Manning's n is changed within an acceptable limit to bridge the gap between observed and simulated
water levels. Manning's M (reciprocal to Manning's n) is used in the model to specify the bed
resistance. In the present study, the Manning number for the Brisbane River-floodplains and
Moreton Bay is used as (M= 10–38 $m^{1/3}$/s) which was based on literature values from previous
modelling calibration of Brisbane River (X. Liu & Lim, 2017; Yu, 2017) and Moreton Bay (Barton
et al., 2017). The Smagorinsky factor of eddy viscosity (Cs = 0.28) based on the literature review
(DHI, 2017b; Li et al., 2020) was adopted to perform the hydrodynamic simulation and model
validation. Model calibration and validation are described in detail in section 4.
*3.6    Model performance evaluation indices*
To compare the observed and simulated results various statistic methods were used e.g. Nash-
Sutcliffe model efficiency coefficient (E), Root Mean Square Error (RMSE) and Coefficient of
determination ($R^2$). Nash-Sutcliffe coefficient (E) was used to assess MIKE 21 model predictive
power and to describe quantitatively the accuracy of simulation results with the observed values. It is
defined in Eq. 8:

$$E = 1 - \frac{\sum_{i=1}^{n}(X_{obs,i} - X_{model})^2}{\sum_{i=1}^{n}(X_{obs,i} - \overline{X_{obs}})^2} \tag{8}$$

Where $X_{model}$ is simulated values at the time *i* and $X_{obs}$ are observed values. Primarily, model efficiency
close to 1, represent accurate results. Root mean square error (RMSE) was used to measure the
difference between simulated values by a model and the observed values, it is defined in Eq. 9:





$$RMSE = \sqrt{\frac{\sum_{i=1}^{n}(X_{obs,i} - X_{model,i})^2}{n}} \qquad (9)$$

Where, $X_{model}$ is simulated values at the time $i$, $X_{obs}$ is observed values

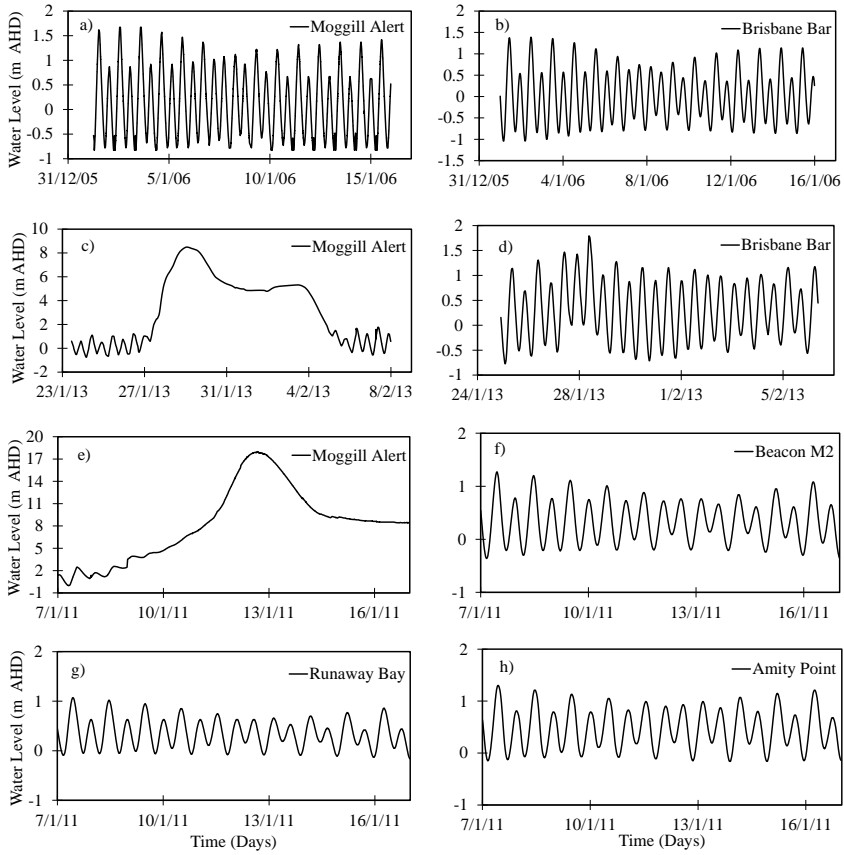


Fig. 6: MIKE 21 model input boundary water level and tidal data

## 4    Results and discussion

*4.1    Calibration and validation of the model*

A comparison between simulated and observed water levels at Brisbane City gauge for the low flow
event confined inside the Brisbane River during the year 2006 is shown in Fig. 7a. For the year 2013
high flow event, spreading over floodplains, the comparison between observed and simulated water





level at Brisbane City gauge is shown in Fig. 7b. The hydrodynamic model is calibrated against the
low flow event 2006, 2013 and to increase the predictive power it is validated for the high flow
event of 2011. For the year 2011 flood event, extending over floodplains, the simulated results are
compared with those observed at Brisbane City and Brisbane Bar gauge as shown in Fig. 7 (c&d).
The performance indices for calibrating gauging stations are shown in Table 4. The statistical results
showed that $E_{ns}$ of water level's varied from 0.84 to 0.95, RMSE values are less than 0.3 and $R^2$
values varied from 0.85 to 0.96, which shows a good match between the observed and simulated
water levels.
Table 4: Performance indices for the calibration and validation of gauging stations of the 2006, 2013
and 2011 flow events

| Year | Gauging stations | RMSE | $E_{ns}$ | $R^2$ |
|------|------------------|------|----------|-------|
| 2006 |                  | 0.25 | 0.84 | 0.85 |
| 2013 | Brisbane City    | 0.2  | 0.87 | 0.88 |
| 2011 |                  | 0.3  | 0.94 | 0.95 |
|      | Brisbane Bar     | 0.1  | 0.95 | 0.96 |

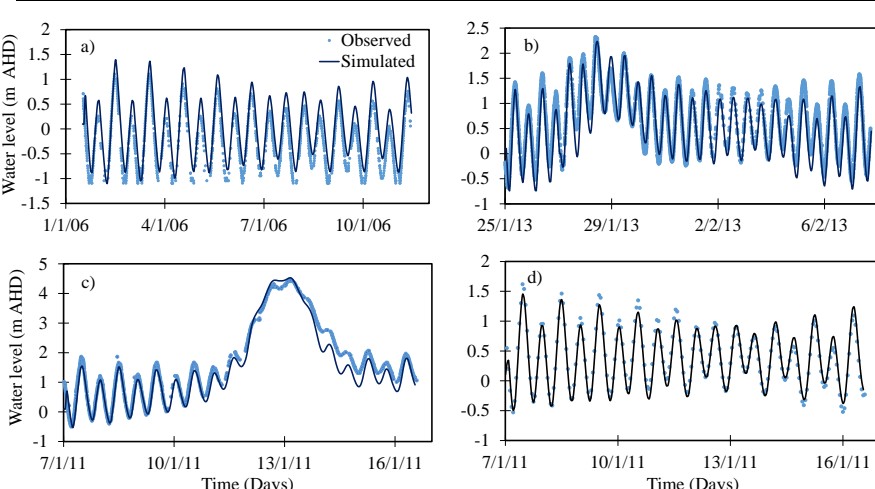


Fig. 7: Observed and simulated water levels at a) Brisbane City for the year 2006; b) Brisbane City
for the year 2013; c) Brisbane city for the year 2011 and d) Brisbane Bar for the year 2011.





The result of the comparison between observed and simulated flood extent is shown in Fig. 8.
Brisbane City experienced a major flood from 12[th] January 10:00 am to 13[th] January 6:00 pm of 32
hours duration (van den Honert & McAneney, 2011). The observed flood extent on 13[th] January
2011 at 04:00 am compared well with the simulated flood extent. The model predictions of flood
extent are 90% accurate while using mesh case 5, which are substantially improved as compared to
(X. Liu & Lim, 2017) with 66.9 % accuracy. The model correctly regenerated most of the Oxley
Creek floodplain and the largest areas of observed flooding below the Jindalee floodplain (Fig. 8),
due to the correct representation of bathymetry and boundary data which leads to correct flood
extend assessment as discussed by (Shrestha et al., 2020). However, the model underestimated the
flood extent in very small tributaries adjoining the Brisbane River, due to the lack of a very finer
mesh size in these areas, because it would lead to an increase in computational time. Based on the
model's good representation for the majority of the floodplain areas, the results can be used for
future predictions.

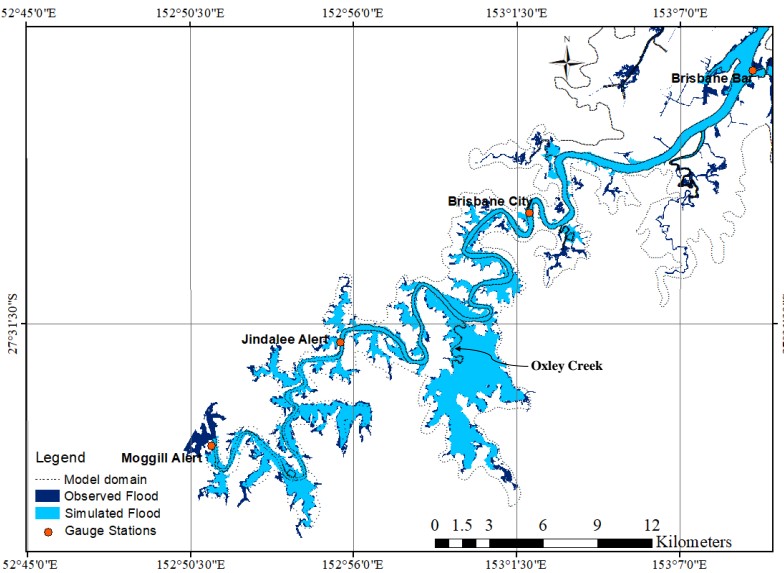


Fig. 8: Simulated and observed inundation area of Brisbane catchment for 2011 flood event





*4.2    Mesh resolution effects on discharge*
The model performance results for the simulated discharge by using different mesh resolutions at
Brisbane City gauge is shown in Fig. 9. The results display a higher difference of coarser mesh with
the observed data and as mesh size become finer the observed and simulated discharges reduce,
indicating that the simulated discharges were correctly represented by the finer mesh resolution, as
also proposed by (Teng et al., 2017). The percentage difference of peak value of simulated
discharges with estimated discharge by Barton et al. (2017) was 16.54%, 14%, 12.19%, 2.88% and
2.7% from Case 1 to Case 5 respectively. The decreasing difference with observed values showed
that the quality of simulation results was gradually enhanced by refining the mesh size; conversely,
further decreases in mesh size create comparatively less difference from cases 4 to 5. Further, it was
found that with coarser mesh size, the hydrodynamics features (i.e. current velocity and discharge)
of the BRE might not be reproduced in the simulation, however with finer mesh size the
performance of the model was enhanced which agreed with the findings of other studies (Shrestha
et al., 2020).

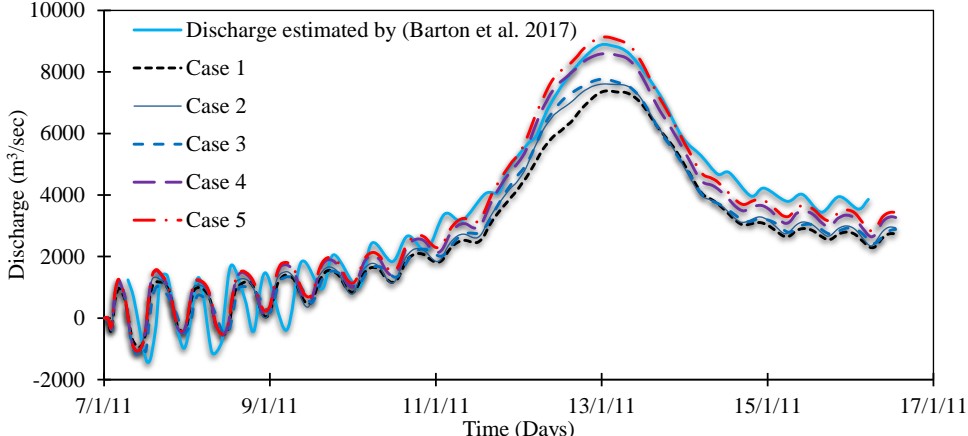


Fig. 9: Comparison of simulated discharge at Brisbane City gauge with estimated discharge by
(Barton et al., 2017) for various mesh resolution cases





### 4.3 *Modelled water levels and flood extents under varying boundaries*

The results of the interaction of storm-tide and fluvial flooding mechanisms by modelling a compound flooding event in the BRE are shown in Fig. 10. The simulated and observed water levels with and without river and tidal boundaries are presented at three gauge locations in Fig. 10. The flood extents corresponding to these boundaries are presented in Fig. 11. Comparison of results at the Jindalee gauge station (Fig. 10a) with and without tidal boundaries show that the peak water level varied slightly, with a 0.62 m reduction at peak level without tidal input. Further, without tidal boundary, the hydrograph has attained smooth rising and falling limbs without showing any tidal variations. While, without discharge boundary i.e. Q=0, the tidal input moved up to the Jindalee gauge, and caused a slight reduction in tidal levels. The comparison of peak water level with and without tidal boundary at Brisbane City gauge (Fig. 10b) shows that the difference of peak flood level could be as high as 0.12 m, while without riverine boundary the water level followed the tidal wave pattern at Brisbane City gauge. At Brisbane bar, without tidal boundary, the water level followed a straight line, with a slight increase in water level during the flood days, while the tidal level at Brisbane Bar was slightly reduced without riverine boundary (Fig. 10c).

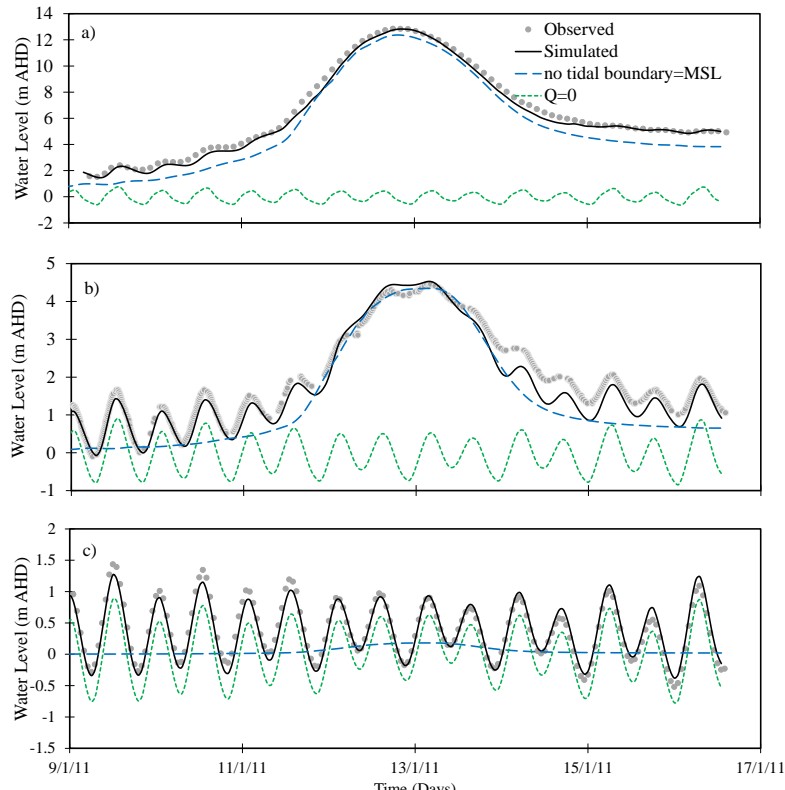

Fig. 10: Modelled and observed water levels for different boundaries (i.e. without tidal boundary = mean sea level (MSL); without discharge boundary, i.e., Q=0) at three gauges; a) Jindalee; b) Brisbane City and; c) Brisbane Bar

Modelled flood extents show a great difference between the two modelling set-ups. The spatial differences of flood extent resulting from simulations with and without river discharge are shown in Fig. 11. The flood extent resulting from the combination of river discharge and tidal water levels are shown as lighter blue areas in Fig. 11, whereas dark blue areas show the flood extent resulting only from tidal water inputs. The comparison of simulated flood extents resulting from these simulations shows variation in the lower BRE floodplain. The inclusion of river discharge caused substantial flood extent within the Oxley creek floodplain, while without river discharge the tidal





water level was confined within the BRE and caused flood extent in floodplain areas.

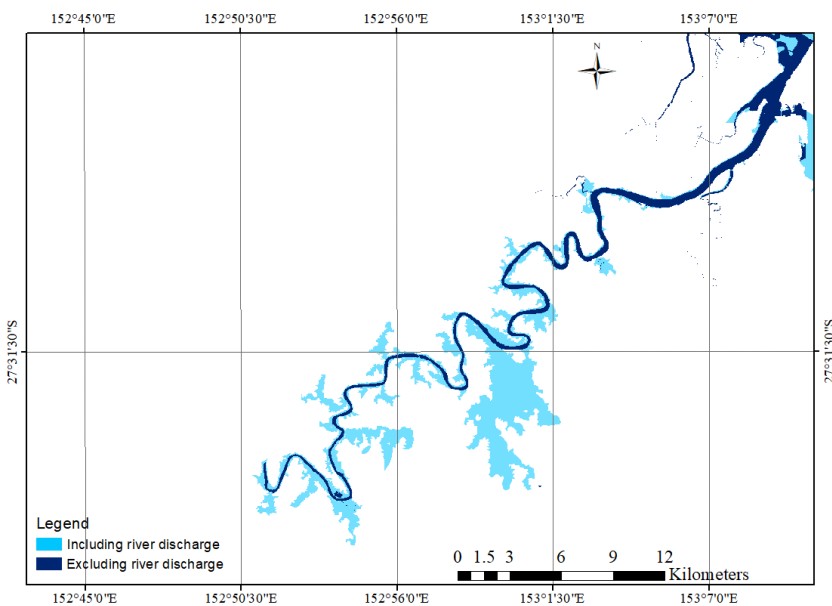


Fig. 11. Simulated maximum flood extents at Brisbane catchment for 2011 flood with and without
river discharge using the two open boundary model setup.
*4.4   Modelled flood extents under future storm surge cases*
Simulated results of flood extent in BRE based on four storm surge scenarios (Fig. 4) are presented
in Fig. 12.  The normal tide flows inside the BRE, without causing any flooding. For future Scenario
1, with a 25% increase in tidal level, the tidal input mainly remained inside the BRE, while causing
very minor flooding near the estuary mouth (Fig. 12 a). The tidal level inside the BRE was just below
the minor flood level of 1.7 m. In Scenario 2, with a 50% increase in tidal level, the tidal level crossed
the minor flood level and tidal inflow caused flooding in the tributaries adjoining the BRE (Fig. 12b).
However, with Scenarios 3 and 4, the flood extent increased near the BRE mouth and Brisbane City
gauge area (Fig. 12 c&d). Further, the flood water level surpassed the minor flood level and levelled
with a medium flood level of 2.6 m in Scenario 3 and 4 respectively.

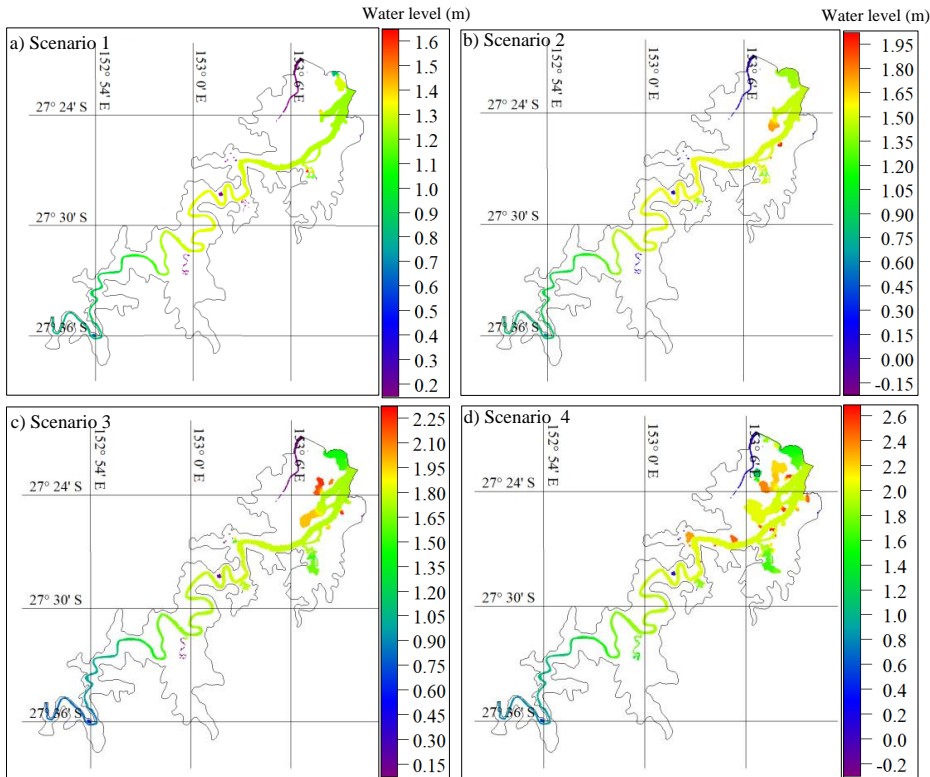

Fig. 12. Simulated flood extents of Brisbane catchment of four scenarios of storm surge
Further, the storm surge scenario 4 was analyzed, with the 2011 and 2013 floods in BRE and the
flood extents are demonstrated in Fig. 13. The floodwater level during the 2011 flood at Brisbane city
gauge was observed as 4.46 m, while considering the future storm surge scenario the flood water at
Brisbane City gauge increased to 5 m due to the joint probability of riverine and tidal effects and
hence flooding extent and depth increase in the floodplain area, with more flooding at the BRE mouth
(Fig. 13 a). Similarly, the flood height for the 2013 flood increased from 2.24 m to 3.01 m due to the
joint probability of river and storm surge, crossing the medium flood level of 2.6 m at Brisbane city
gauge and leading to flooding in the Oxley creek area (Fig. 13 b). The modelling with the future storm
surge scenarios has shown that for flood inundation study and coastal planning in BRE, the
combination of riverine flow and the storm surge effect due to climate change were considered. As


with the increasing tidal height, the tidal impact pushed low river flow, while moving along the BRE
beyond Moggill gauge, and causing minor and moderate flooding.

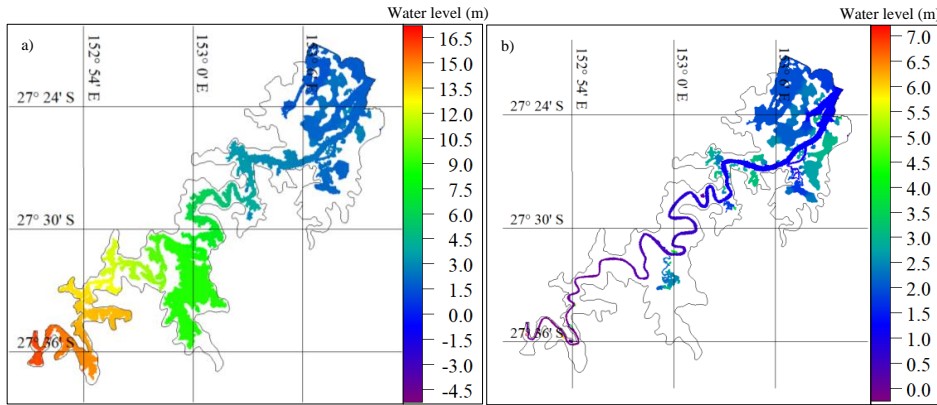

Fig. 13. Simulated flood extents of Brisbane catchment; a) 2011 and; b) 2013 flood events in
combination with future storm surge scenario 4
**5    Conclusions**
To simulate flood height and extent in Brisbane River Estuary, mesh resolutions and combined flood
effect by using the MIKE 21 Model were analysed. MIKE 21 hydrodynamic model was calibrated
and validated for the years 2006, 2013 and 2011 flow events, with a flow Nash–Sutcliffe coefficient
(E) between 0.84 and 0.95 at all gauges. The model simulated the flooding extent of BRE showing
more than 90% accuracy. This confirmed that the MIKE 21 model can dynamically simulate and
replicate the flows with a compound flood event.

Five mesh resolutions cases were analyzed and the result found that finer mesh resolution produces
more   accurate   results   and   performs   better   with   hydrodynamic   features.   Moreover,   model
performance evaluation showed that the mesh structure in Case 5 was more appropriate than the
others, considering the convergence tendency of the simulation results and running time.



Compound flood event simulation results emphasized that not considering the interaction of various
flooding drivers have caused 0.62 m and 0.12 m reduction in flood levels at Jindalee and Brisbane
City gauges while leading to substantial underestimation of flood extent. Results endorse the
consideration of tidal and riverine flooding drivers mutually for coastal flood extents assessments
in estuaries.
Simulated results of flood extent in BRE based on four storm surge scenarios show that the flooding
level would cross the medium flood level at Brisbane City gauge. Further with 2011 and 2013 floods
with storm surge scenario 4, it was demonstrated, that the flood level will increase to 12% and 34%
respectively and flood extent will increase in Oxley creek and near the BRE mouth.
The results show that a flood hydrodynamics study of BRE using compound flood events and
considering future storm surge analysis would be helpful for coastal managers for planning and
management of coastal projects.
**Author's contribution**
U.K.: Conceptualization, methodology, writing—original draft, review editing and investigation.
S.Y.: Conceptualization, review, editing, and supervision. M.S (Muttucumaru Sivakumar).
Review editing and supervision. K.E.: Review, editing and supervision. M.S (Mariam Sajid):
Investigation, writing, review and editing. M.Z.B.R.: Review and editing. All authors have read
and agreed to the published version of the manuscript.
**Declaration of Competing Interest**
The authors declare that they have no known competing financial interests or personal
relationships that could have appeared to influence the work reported in this paper.
**Funding:** This research received no external funding
**Acknowledgments:** The authors would like to thank the Queensland Government Departments
for providing all necessary input data. Special thanks are to Daryl Metters (Department of
Environment and Science), Jim Fear (SEQ Water), Paul Boswood (Department of Environment
and Science), Paul Birch (Bureau of Meteorology) and Paul Finger (Maritime Safety Queensland).





We are thankful to DHI for providing the MIKE 21 license. The authors would also like to thank
Caroline Lai and Méven Robin Huiban for providing the technical support for MIKE 21.
**Conflicts of Interest:** The authors declare no conflict of interest.

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
