# Peer review of "Modelling the compound flood hydrodynamics under mesh convergence and future storm surge events in Brisbane River Estuary, Australia"

_Natural Hazards and Earth System Sciences, 2021_

## Referee Comment (RC1)

**Review of "Modelling the compound flood hydrodynamics under mesh convergence and future storm surge events in Brisbane River Estuary, Australia " (submitted to NHESS by Usman Khalil, Shuqing Yang, Muttucumaru Sivakumar, Keith Enever, Muhammad Zain Bin Riaz, Mariam Sajid)**

The authors develop a Brisbane River estuary and Moreton Bay flood model for various flows events under converging mesh size. A simulation to assess the impact of compound riverine flooding and tides on water levels and analyze the future storm surge effect on flood extent.

Criteria:
1) Scientific Significance Does the manuscript represent a substantial contribution to the understanding of natural hazards and their consequences (new concepts, ideas, methods, or data)?

The idea of the manuscript is interesting, and the datasets and methods used are appropriate.
Yes, the study has enough scientific merit for publication. It is not particularly original, but is a useful case study.

Excellent     Good   Fair   Poor

2) Scientific Quality
Are the scientific and/or technical approaches and the applied methods valid?
Are the results discussed in an appropriate and balanced way (clarity of concepts and discussion, consideration of related work, including appropriate references)?

The abstract is good, although a little editing would make it even better. The logic followed in the abstract and introduction is clear. However, the methodology is poorly described. It is necessary to establish in Mike 21 model calibration how the different adjustment parameters influence the modeling conditions for each case in order to be able to perform traceability of results (you could add a table for each case). It is also necessary to incorporate the errors of the parameters as they influence the scales of the intervening processes.
The manuscript's results are presented in a way that allows the reader to draw their own conclusions.

The manuscript is quite short. There is enough space for the authors to expand their introduction, methodology and conclusion considerably, with more discussion….

Excellent     Good   Fair   Poor

3) Presentation Quality
Are the scientific data, results and conclusions presented in a clear, concise, and well-structured way (number and quality of figures/tables, appropriate use of technical and English language, simplicity of the language)?

Possibly – the figures could be clarified and quality too, please standardize font sizes in graphics.  The English language is clear…

Excellent     Good   Fair   Poor

The manuscript is accepted with minor corrections.

---

## Referee Comment (RC2)

Review of manuscript

"Modelling the compound flood hydrodynamics under mesh convergence and future storm surge events in Brisbane River Estuary, Australia" by Usman Khalil, et al.

General comments:

This manuscript investigates the optimal mesh resolution of the MIKE 21 model to simulate the river discharge. Furthermore, authors are attempting to analyze the co-variance of flooding from the sea and rivers by modeling the storm surges and fluvial floods jointly. As compound events become more of a problem because of climate change, it may provide some guidance for future risk management of coastal flooding.

I discovered that the manuscript is quite intuitive. Previous research employed MIKE 11 to simulate fluvial flooding while ignoring the sea level from the sea side (storm surge), or MIKE 21 to simulate storm surge while ignoring the water level at the river mouth. As a result, some research into this new direction is highly recommended. However, the results of these simulations do not satisfy me. The finding is plausible, and one might reach the same conclusion qualitatively without using any model simulations, for example, the finer the mesh resolution, the more accurate it is compared to the observation. I'd rather assume that the resolution in Case4 is sufficient, as the difference between 2.88 % and 2.7 % is not significant (Line 329). Do the simulations provide enough improvement in terms of quantification to make a risk management recommendation? For example, what means the inaccuracy of the peak discharge in finer resolution is 2.7 % and what's the impact of 0.62m reduction in sea level without considering the compound effects? I believe the authors should devote more time to such quantifications and extract the information that is beneficial for risk management. Furthermore, some of the basic concepts in risk management, like exposure and vulnerability, should be introduced and discussed, if authors believe this would be the major contribution to this field.

I propose that authors distinguish between storm surge and mean sea level rise in the future. I've seen authors combine those two topics. I would rather refer to future mean sea level rise scenarios for the experiments in this manuscript. Alternatively, authors should clarify or demonstrate that mean sea level increase is the primary contributor to future storm surge (extreme sea level) changes.

Minor comments:

**Title:** Is 'mesh convergence' a well known term? I understood there are two directions (mesh resolution, and compound events in future) in this manuscript, but I believe there should be a 'meeting point', and reflected in title.

**Abstract**

L16 , 'investigate the flooding processes …' is it from sea side?

L20, There is a 'jump' here. I do not believe N-S coefficient is well known parameter.

L23, 'with 2.7% with estimated discharge'? It sounds like the error is between peak discharge and estimated discharge. It should be the error between simulated and estimated discharge during peak discharge time, right?

L26, 'and uncertainties in flood extent' could be add as a plus-minus sign to 0.62 and 0.12

**Introduction**

L40, 'USD'should be '$US', to be consistent with L38?

L56, I think here could start a separate paragraph for modeling work. The first paragraph is quite long.

L82, 'The recent flooding' is the right word to describe such a widespread disaster.

L85-86, I believe only events could not 'specify the needs'. I would more believe, 'more frequent or more costly events' is the wording should be specified here.

L92-93, 'tidal flooding' is not defined. Is it should be 'storm surge' as defined.

**Methods**

L221-223, this key part of the 'mesh convergence' is not introduced clearly. As a concept of 'convergence', one needs to present an asymptotic line and define a threshold. Furthermore, convergence is not only related to the error of the water levels, it should be combined with the computation time together (in L218).

Fig. 3, very bad quality figures.

L234-243, should be moved to Introduction?

Fig.5, base scenario and scenario4 have very similar blue color.

L262, if MIKE 21 has not considered wind and pressure, the application of this model in a real storm surge case is very limited. In principle, it can be only used for some numerical experiments as what authors did. So some discussion on the direction is needed.

**Results**

L316-318, I can understand this statement. For sure, better resolution gives better results. However, there is no quantification in the increase in computational time. I could not see how authors considered the trade-off between resolution and computational costs. Therefore, a bit more details (maybe on why errors of flood extents in these areas are rather acceptable) would be very helpful for the similar applications.

Fig.12 I have not seen sewage system is mentioned. I have seen some papers discussed the impact on sewage system when considering different flood extents scenarios. Is it relevant to Brisbane?